# Improvement of *Pseudoalteromonas haloplanktis* TAC125 as a Cell Factory: IPTG-Inducible Plasmid Construction and Strain Engineering

**DOI:** 10.3390/microorganisms8101466

**Published:** 2020-09-24

**Authors:** Andrea Colarusso, Concetta Lauro, Marzia Calvanese, Ermenegilda Parrilli, Maria Luisa Tutino

**Affiliations:** Dipartimento di Scienze Chimiche, Complesso Universitario Monte Sant’Angleo, Via Cintia, 80126 Napoli, Italy; andrea.colarusso@unina.it (A.C.); concetta.lauro@unina.it (C.L.); marzia.calvanese@unina.it (M.C.); erparril@unina.it (E.P.)

**Keywords:** *Pseudoalteromonas haloplanktis*, strain engineering, Lon protease, *Ec*LacY, recombinant protein production, IPTG, pP79 vector

## Abstract

Our group has used the marine bacterium *Pseudoalteromonas haloplanktis* TAC125 (*Ph*TAC125) as a platform for the successful recombinant production of “difficult” proteins, including eukaryotic proteins, at low temperatures. However, there is still room for improvement both in the refinement of *Ph*TAC125 expression plasmids and in the bacterium’s intrinsic ability to accumulate and handle heterologous products. Here, we present an integrated approach of plasmid design and strain engineering finalized to increment the recombinant expression and optimize the inducer uptake in *Ph*TAC125. To this aim, we developed the IPTG-inducible plasmid pP79 and an engineered *Ph*TAC125 strain called KrPL *LacY^+^*. This mutant was designed to express the *E. coli* lactose permease and to produce only a truncated version of the endogenous Lon protease through an integration-deletion strategy. In the wild-type strain, pP79 assured a significantly better production of two reporters in comparison to the most recent expression vector employed in *Ph*TAC125. Nevertheless, the use of KrPL *LacY^+^* was crucial to achieving satisfying production levels using reasonable IPTG concentrations, even at 0 °C. Both the wild-type and the mutant recombinant strains are characterized by an average graded response upon IPTG induction and they will find different future applications depending on the desired levels of expression.

## 1. Introduction

Over recent years, both constitutive promoters [1] and inducible cassettes [2,3] have been established for the recombinant expression in *Pseudoalteromonas haloplanktis* TAC125 (*Ph*TAC125) in a wide range of temperatures. Regulatable systems are particularly desirable in industrial processes where the decoupling of the biomass accumulation from the recombinant expression could be crucial to guarantee satisfactory yields. Although they proved to be useful for a series of studies [2,3,4,5,6], the two inducible expression vectors used in *Ph*TAC125 so far showed some major drawbacks. The L-malate inducible pUCRP plasmid guaranteed a remarkable protein accumulation [2], but its efficacy resulted in be strongly influenced by the medium composition. In particular, the use of L-glutamate as carbon source negatively affected pUCRP induction, making it necessary to formulate bacterial media devoid of this amino acid [5]. Given the pivotal contribution of such a carbon source to *Ph*TAC125 specific growth rate and metabolic regulation [7], its depletion might limit the versatility of this recombinant system for industrial purposes. On the other hand, the D-galactose regulatable pMAV expression vector showed a good versatility in terms of the temperature range of use when *P. haloplanktis* TAE79 β-galactosidase was employed as a reporter [3]. Nevertheless, the amount of enzyme that could be accumulated in recombinant *Ph*TAC125 pMAV-*lacZ* was lower than the yield achievable in the nonrecombinant parental *Ph*TAE79 strain [8], suggesting a low strength of the used inducible promoter.

The β-galactosidase production in *Ph*TAE79 wt was indeed sufficiently high to guarantee its purification from *Ph*TAE79 extracts and its industrial exploitation for lactose treatment without the use of any recombinant technology [8,9]. This data induced us to evaluate the potential translatability of the regulatory sequences of *Ph*TAE79 *lacZ* in *Ph*TAC125 for recombinant purposes. The preferable choice of *Ph*TAC125 rather than *Ph*TAE79 as a host descends from the wider available information in terms of genomic organization and annotation [10], genetic modification strategies [11], and metabolic networks [5,7,12,13] for the first bacterium.

Based on an in silico analysis of the *Ph*TAE79 *lacZ* expression cassette and on previously published data from other authors, we were persuaded of the feasibility of the use of this regulatory system to develop a new expression vector in *Ph*TAC125. This plasmid, called pP79, proved to be IPTG-inducible, to outperform our previous regulated expression vector pMAV, and to allow the detection of the production of a fluorescent protein in *Ph*TAC125 for the first time. To better the performance of this system, we genetically engineered the host for the expression of a mesophilic lactose permease. Such a mutant strain guaranteed a higher recombinant production using a lower IPTG concentration range in comparison with the parental strain. Collectively, our results emphasize the remarkable flexibility of *Ph*TAC125, a polar host capable of combining a heterologous psychrophilic expression system with a mesophilic inducer transporter for the recombinant production of proteins.

## 2. Materials and Methods 

### 2.1. Bacterial Strains and Growth Media Formulations

The strains used in this study are listed in Appendix A. *E. coli* DH5α was used for cloning procedures, while *E. coli* S17-1(λ*pir*) was employed in intergeneric conjugations as a donor strain for KrPL transformations [14]. KrPL—a cured *Ph*TAC125 strain—was used in all the recombinant expression and mutagenesis experiments. *E. coli* was cultured in LB broth (10 g/L bacto-tryptone, 5 g/L yeast extract, 10 g/L NaCl) at 37 °C and the recombinant strains were treated with either 34 µg/mL chloramphenicol or 100 µg/mL ampicillin, depending on the selection marker of the vector. KrPL was grown in TYP (16 g/L bacto-tryptone, 16 g/L yeast extract, 10 g/L NaCl) during conjugations and precultures development, and in GG [3] in expression growths. For the propagation and culture of KrPL recombinant strains, either chloramphenicol or ampicillin was used. In detail, chloramphenicol was added to solid and liquid media at 12.5 µg/mL and 25 µg/mL concentrations, respectively. Ampicillin was always used with a concentration of 100 µg/mL, instead.

### 2.2. Construction of pP79 and p79C Expression Plasmids

The AUTL01000130.1 contig containing *Ph*TAE79 *lacZ* (Figure 1) was automatically annotated with RAST [15] and the annotations were refined using BlastP [16]. For all digestion/ligation reactions, NEB enzymes were used (New England Biolabs, Hitchin, UK). The restriction sites in pP79 and p79C that were hydrolyzed for cloning purposes are visible in the maps in Figure 2. The pP79 inducible expression vector was designed by cloning the DNA fragment from *Ph*TAE79 encompassing the *lacR* gene and the divergent *lacZ* promoter + 5′ UTR into pUCLT/Rterm vector [2,17]. To this aim the pSP73-*β-gal* vector [8] was used as a template in a PCR involving the use of p79_fw and p79_rv primers (Appendix A). The resulting ~1.2 kb amplified sequence covered the 18′782–19′909 region of AUTL01000130.1 and was characterized by the addition of SphI restriction site to its 5′ extremity and NdeI, SalI and XbaI to its 3′ terminus. Both pUCLT/Rterm and the p79 amplicon were double digested with SphI/XbaI and ligated. The resulting plasmid was pP79.

p79C is a variant of pP79 harboring a chloramphenicol resistance marker rather than an ampicillin selection gene. For its construction, pP79 regulatory sequences, its MCS and the *aspC* transcriptional terminator were extracted with SphI and SacI from pP79 and ligated with pUCC [4] digested in the same sites.

### 2.3. Sub-Cloning of Heterologous Genes into the Expression Plasmids

The expression plasmids used in this work are reported in Appendix A. pMAV-*lacZ* was prepared in a previous study [3] and was used to isolate the psychrophilic β-galactosidase encoding gene for the construction of pP79-*lacZ.* In particular, *lacZ* was split into two different fragments: one of 1.2 kb with NdeI/NcoI extremities and the second of 2.3 kb with NcoI/XbaI extremities. The two gene fragments were ligated with pP79 opened with NdeI and XbaI restriction sites. For the conversion of pP79-*lacZ* into p79C-*lacZ*, the same approach described in Section 2.2 was employed: the regulatory *lacR* gene and *lacZ* were isolated using SphI/SacI double digestion and inserted into pUCC hydrolyzed with the same enzymes.

The *R9-gfp* gene was taken from pET-21b-*R9-gfp* [18] using NdeI and HindIII restriction sites. In the detail, the HindIII digestion was performed first and then the extremities of the hydrolyzed vector were filled by Klenow reaction. After NdeI digestion, the *R9-gfp* gene was cloned into pMAV with NdeI/filled-EcoRI extremities. pMAV-*R9-gfp* was then converted into pP79-*R9-gfp* by replacing pMAV typical expression sequences with the ones of pP79. To do so, ScaI/NdeI double digestion was used to isolate the pP79 fragment encompassing its promoter and its regulatory gene (2.1 kb). This was cloned into the pMAV-*R9-gfp* backbone devoid of the *gal*T expression sequences isolated with the same restriction sites (3.6 kb). pP79-*pGFP* was designed to drive the expression of a codon optimized version of the eGFP [19]. Composition optimization of the *pGFP* for the codon usage of *Ph*TAC125 was automatically performed with the Optimizer web tool using the “guided random” method [20]. The synthesized gene (Thermo Fisher Scientific, Waltham, MA, USA) was cloned into pP79 using NdeI and KpnI restriction sites.

p13C-*lacY* and pFC-*lacY* were the two constructs used for the constitutive expression of *E. coli lacY* gene. Briefly, the gene encoding the mesophilic lactose permease was synthesized by Thermo Fisher Scientific (Waltham, MA, USA) following a sequence optimization for the codon usage of *Ph*TAC125 [20] and adding a c-*myc* encoding sequence at its 3′ extremity. The insert harbored NdeI and KpnI restriction sites at its 5′ and 3′ ends, respectively, and was cloned into p13C and pFC vectors using the same sites. p13C is a plasmid containing the P13 promoter and a chloramphenicol resistance gene. It was built by fusing P13 sequence taken from pPM13 [1] using HindIII/XbaI double digestion with pUCC [4] hydrolyzed with the same enzymes. pFC contains the constitutive promoter of the *Ph*TAC125 *aspC* gene and was already available [21].

The complete sequences of genes introduced in this study are reported in the Appendix B and Appendix A.

### 2.4. Preparation of pVS-lon and pVS-lacY Suicide Vectors

For the construction of *lon* mutant, two DNA fragments of *Ph*TAC125 *lon* gene (A and B) were amplified by PCR using bacterial genomic DNA as the template. Two primer pairs were designed to amplify a 305 bp region at the 5′ end (lonA_SphI fw, lonA_SacI rv) and a 233 bp region at the 3′ end (lonB_SacI fw, lonB_EcoRI rv) of *lon* gene. The obtained amplicons were subjected to SphI/SacI and SacI/EcoRI double digestions respectively and cloned into the pVS [22] previously digested with SphI and EcoRI, resulting in pVS-*lon* vector.

The construction of pVS-*lacY* was performed starting from the recovery of the fragment P13-*lacY* from p13C-*lacY* vector through hydrolysis with HindIII and KpnI. Then two fragments (B and B’) at the 3′ end of *Ph*TAC125 *lon* gene were amplified by PCR. The reactions were carried out using the genomic DNA as the template and allowed the amplification of a fragment of 233 bp (lonB_SphI fw, lonB_HindIII rv) and one of a 170 bp region (lonB’_HindIII fw, lonB’_EcoRI rv). SphI/HindIII and KpnI/EcoRI double digestions were performed on the obtained amplicons, respectively. The fragment B carrying SphI/HindIII extremities and P13-*lacY* hydrolyzed with HindIII/KpnI were cloned into pUCC vector, previously digested with SphI and KpnI. Afterward, the obtained intermediate vector pUCC-*lonB*-P13-*lacY* was digested with SphI and KpnI to extract the fragment lonB-P13*lacY*. This fragment was finally cloned together with the second amplicon B’ hydrolyzed KpnI/EcoRI into pVS adequately digested with SphI and EcoRI, resulting in pVS-*lacY*.

### 2.5. Transformation of KrPL and Selection of the lon and lacY^+^ Mutant Strains

The recombinant vectors were mobilized into KrPL by intergeneric conjugation [14]. The selection of recombinant transconjugants was performed at 15 °C in the presence of 50 μg/mL kanamycin and either 100 μg/mL ampicillin or 12.5 μg/mL chloramphenicol, depending on the specifically employed vector. 

As for the mutant strains, the transconjugants were selected at 15 °C in the presence of 50 μg/mL kanamycin and 30 μg/mL carbenicillin.

### 2.6. gDNA Extraction from the Mutant Strains and Sequence Analysis

Genomic DNA extraction from KrPL mutants and *Ph*TAE79 was performed using the Bacterial DNA kit (D3350-02, E.Z.N.A™, OMEGA bio-tek, Norcross, GA, USA) following the manufacturer’s instructions. The insertion of the suicide vectors into the cells was verified by PCR analysis with a NEB Taq DNA polymerase (New England Biolabs, Hitchin, UK). The genomic DNA was used as the template of the reactions and two couples of primers were used for the amplification of amp(R) (bla_fw, bla_rv) and pheS (pheS_fw, pheS_rv) genes. Then, further PCR analysis was performed to identify the insertion site into *lon* gene. The couples of primers used for this purpose are:lonA_SphI fw, lon_rv and lon_fw, lonB_EcoRI rv, for the analysis of *lon* mutants;lonY_fw, lacY_rv and lacY_fw, lonY_rv, for the analysis of *lacY^+^* mutants.

### 2.7. Recombinant Production of the Reporter Proteins

Glycerol stocks (−80 °C) of KrPL recombinant strains were streaked over TYP agar selective plates. After three-five days of incubation at 15 °C, a single colony was inoculated in 2–3 mL of TYP at 15 °C for one day. To grow the bacteria in GG, they were routinely trained in the same medium with two subsequent 1/100 dilutions within a time frame of 24 h. The actual inoculum was generally performed in the liquid medium filling an Erlenmeyer flask by 20% of its volume and with a starting OD_600_ of 0.1. For recombinant expression at 15 °C, the cells were generally induced in late exponential phase (OD_600_ = 1) about 13 h after the initial dilution. Strains harboring pMAV derived vectors were induced with 10 mM D-galactose, while pP79 and p79C carrying strains were treated with different concentrations of either IPTG or lactose. Expression trials were attempted also at 0 °C in a similar way as described above. In this case, Erlenmeyer flasks were filled by 35% of their volume to stem oxidative stress and the growths lasted several days considering that KrPL generation time was about 24 h at 0 °C. In most of the experiments, a Biosan PSU-20i orbital shaker was used setting the agitation at 180–220 rpm.

### 2.8. Analysis of the Production of the Recombinant Proteins

For the analysis of the β-galactosidase production, 10 OD_600_ pellets were harvested during the cultures by centrifugation (5000× *g* for 5 min at 4 °C) and resuspended in 0.4 mL of Lysis buffer (100 mM sodium phosphate buffer pH 7.5, 2% (*v/v*) Triton X-100, 1 mM DTT, 5 mg/mL lysozyme). After 20 min of incubation at 15 °C, the samples were centrifuged (10,000× *g*, 15 min, 4 °C) and the supernatants were used in the following enzymatic measurements with ONPG as a substrate. The spectrophotometric assays were performed in triplicate as reported by Hoyoux et al. [8] and the data analysis was carried out using the ONPG extinction coefficient at 410 nm (3.5 mM^−1^ cm^−1^) and the total protein concentration measured with the Bradford assay (Bio-Rad Laboratories, Hercules, CA, USA).

To monitor the production of fluorescent proteins, 1 OD_600_ of liquid cultures was centrifuged at 5000× *g* for 5 min at 4 °C and the pellets were resuspended in 0.5 mL PBS. Then, the samples were serially diluted to achieve the best signal to noise ratio in fluorescence measurements and the dilution factor was used for normalization. Fluorescence measurements were conducted with a JASCO FP-750 spectrofluorometer at 25 °C with an excitation wavelength of 488 nm (slit 3 nm), an emission wavelength of 509 nm (slit 6 nm) and an integration time of 0.10 s.

The production of the recombinant proteins was also monitored by SDS-PAGE, by loading 20 µg of soluble cellular extracts onto the wells of 10% denaturant gels. The cellular homogenization was carried out through the chemical-enzymatic method indicated at the beginning of this section and the total protein concentration in the soluble fractions was estimated with the Bradford method. In the case of GFP producing strains, we also checked for the synthesis of the recombinant proteins in total lysates to control their solubility. Nevertheless, the presence of R9-GFP and pGFP was never visible both in the soluble and total extracts when run onto SDS-PAGE.

To verify the presence of the truncated form of Lon protease, 1 OD_600_ cell pellets were collected by centrifugation and solubilized in 60 μL of Laemmli buffer 4X. Then, the samples were boiled at 95 °C for 20 min, quickly cooled on ice for 5 min and finally centrifuged at 10,000× *g* for 5 min at RT. 5 μL of samples were analyzed by SDS-PAGE. 4–15% Mini-Protean TGX (Biorad) gels were used in TGS buffer setting the power supply to constant 120 V. For electroblotting, the Biorad Transblot Turbo system with Biorad PVDF mini membranes was used employing the mixed molecular weight setting. After the transfer, the membrane was blocked with PBS, 0.05% Triton X-100, 5% (*w/v*) milk for one hour. Then, an anti-Lon antiserum (ab103809) was diluted 1:1,000 in the same buffer. After one hour of incubation at RT with the primary antibody, the membrane was washed with PBS, 0.05% Triton X-100 three times (5 min each) and incubated with an anti-rabbit antibody diluted 1:30,000 in PBS, 0.05% Triton X-100, 5% (*w/v*) milk for one hour at RT. Then, the membrane was washed again with PBS, 0.05% Triton X-100 three times (5 min each) and the secondary antibody was detected using the ECL method.

### 2.9. mRNA Extraction and qPCR

Total RNA was isolated from the cells using the Direct-zol RNA Kit (Zymo Research, Irvine, CA, USA) following the manufacturer’s instructions. Contaminating genomic DNA was then removed through treatment with RNAse-free DNase I (Roche, Mannheim, Germany). Total RNA was reverse transcribed using SuperScript IV (Invitrogen, Carlsbad, CA, USA) according to the recommended protocol. The primers used for this reaction are listed in Appendix A. Quantitative real-time PCR was performed on cDNA from each sample by using PowerUp SYBR Green Master Mix (Applied Biosystems, Foster City, CA, USA) implemented with the specific primers (listed in Appendix A) in StepOne Real-time PCR System (Applied Biosystems, Foster City, CA, USA). The housekeeping gene *PSHA_RS01090* was chosen as the normalizer. The expression level of the gene of interest was assayed for up-regulation in experimental samples in comparison to a calibrator sample (NI). The relative quantification of mRNA was expressed as fold-change and was calculated through the standard curve method [23]. Three independent sets of experiments were performed.

## 3. Results

### 3.1. Analysis and Cloning of the PhTAE79 lacZ Expression Sequences

The *Ph*TAE79 genome has been sequenced in the framework of a WGS project involving several Antarctic *Pseudoalteromonadales* [24], but it was neither assembled nor annotated. The *lacZ* gene is in the AUTL01000130.1 contig according to the GenBank notation, whose predicted genes distribution has been schematized in Figure 1. In this ~24 kb region, *lacZ* clusters with other genes involved in carbohydrates and amino acids metabolism (Figure 1, blue arrows) and, intriguingly, with three predicted CDSs involved in cut and paste mechanisms (black arrows) and a putative RepB protein involved in plasmid replication (red arrow) [25]. In particular, the RepB is highly conserved (65–95% nucleotide identity) in plasmids harbored by three marine *Pseudoalteromonadales*, *P. haloplanktis* TAC125 (MN400773.1), *P. nigrifaciens* KMM 661 (CP011038.1), *P. arctica* A 37-1-2 (CP011027.1), whose reciprocal similarities have been recently examined [26]. Considering that the whole analyzed contig is almost totally conserved in *P. nigrifaciens* plasmid (88% coverage with 99% identity), it is very likely that the DNA containing the *lacZ* gene is the result of horizontal gene transfer also in *Ph*TAE79.

Upstream and divergent to *Ph*TAE79 *lacZ* is a gene predicted to encode an AraC family transcriptional regulator (one of the two green arrows in Figure 1), which probably regulates the β-galactosidase mRNA synthesis and, for this reason, it will be named LacR from now on. Hoyoux et al. used a combination of lactose and IPTG to induce the production of the β-galactosidase in *Ph*TAE79 and reported that IPTG addition led to an increased protein yield [8]. We confirmed this outcome by inducing *lacZ* expression in *Ph*TAE79 using IPTG as the only inducer molecule (data not shown). This suggests that LacR is probably regulated by this small allolactose analog.

Persuaded by this preliminary data, we developed a shuttle vector, named pP79, containing *Ph*TAE79 *lacR-lacZ* regulatory elements. In detail, a PCR was designed to amplify the LacR CDS with its putative transcriptional terminator and promoter together with the predicted *lacZ* divergent promoter, its 5′ UTR and initial ATG. Then, the amplicon was cloned into the shuttle vector pUCLT/Rterm [2,17], so to have the pP79 plasmid. To make this expression system compatible with other constructs, we also developed its chloramphenicol resistant version, p79C, by ligating pP79 expression cassette with the pUCC vector (Figure 2A) [4]. The transcription start of the *lacZ* gene indicated in red in Figure 2B was identified with a primer extension assay (data not shown).

### 3.2. Quantification of pP79 Activity Using β-Galactosidase and R9-GFP Reporters

#### 3.2.1. Comparison between pP79 and pMAV Efficiencies

To test the usefulness of pP79, we compared its performance with our most recent inducible expression system, pMAV [3]. To this aim, we used two different reporter genes, *Ph*TAE79 *lacZ* that has been employed for the characterization of all the expression plasmids in *Ph*TAC125 so far [1,2,3], and *R9-gfp* which encodes a GFP variant tagged with an N-terminal R9 peptide [18]. In particular, the GFP protein encoded by this construct harbors the eGFP mutations for enhanced fluorescence [19] and the Cycle 3 mutations for improved folding [27]. To ensure plasmids stability, the recombinant constructs were mobilized into KrPL, a *Ph*TAC125 strain cured of its endogenous plasmid pMtBL (unpublished results from this laboratory). The β-galactosidase production in KrPL pMAV-*lacZ* was carried out by D-galactose induction in the defined medium GG at 15 °C [3]. In the case of KrPL pP79-*lacZ* strain, a 1–10 mM IPTG range was tested for induction in the same growth conditions and the levels of accumulated recombinant protein were measured after overnight expression. pP79 proved to guarantee a higher enzymatic specific activity at all the tested inducer concentrations than pMAV, reaching a 20-fold higher production when 10 mM IPTG was used (Figure 3A). This result was confirmed by assessing the fluorescence emitted by R9-GFP producing strains at 15 °C (Figure 3B). When induced with 10 mM D-galactose, pMAV-*R9-gfp* bearing cells had a fluorescence that was at the same level as the autofluorescence of non-recombinant *Ph*TAC125. Conversely, KrPL pP79-*R9-gfp* showed a detectable protein accumulation over time when 10 mM IPTG was added to the culture.

#### 3.2.2. Evaluation of the Reliability of *lacZ* and *R9-gfp* as Reporter Systems

Although the two reporter systems consistently demonstrated that pP79 guarantees a higher recombinant production than pMAV, the fluorescence-based approach suffered from lower sensitivity because pMAV-driven R9-GFP production could not be distinguished from the background noise (Figure 2B). To define if this drawback was due to the intrinsic different sensitivities of the two assays or to a discrepancy in the absolute production of the two reporters, further analyses were carried out. First, we monitored the β-galactosidase productions via SDS-PAGE, which allowed us to detect the presence of the recombinant enzyme both in KrPL pMAV-*lacZ* and KrPL pP79*-lacZ* induced strains (Figure 4A). As expected, even when the lower IPTG concentration of 1 mM was used, the intensity of the estimated 118 kDa band was higher in the cellular extract of KrPL pP79-*lacZ* than the one observable in KrPL pMAV-*lacZ* induced lysate (lanes 2 and 4 in Figure 4A, respectively). However, no protein band was visible in induced KrPL strains producing R9-GFP at the expected molecular weight of 28 kDa, regardless of the employed plasmid and inducer concentration (Lanes 5 and 6 in Figure 4A). To define if this different accumulation of the two recombinant proteins was related to transcriptional issues, *lacZ* and *R9-gfp* mRNAs produced by pP79 were quantified through quantitative real-time PCR. The results were expressed as fold-changes to compare the relative amount of *lacZ* and *R9-gfp* mRNA produced both in the presence and in the absence of IPTG. As reported in Figure 4B, a rapid increase of mRNA and a significant accumulation during the time were observed for both the reporters.

Taken together, these results prove that R9-GFP production is considerably less efficient than the one of the psychrophilic β-galactosidase in *Ph*TAC125 and that this phenomenon is unrelated to transcriptional issues. To determine if the N-terminal polyarginine moiety of R9-GFP and its Cycle 3 mutations were the cause of such a low protein yield, we designed an automatically codon optimized gene encoding the eGFP devoid of any N-terminal tag and sequence mutations other than the ones needed for increased fluorescence. This protein was named pGFP and its fluorescence levels were compared with the ones of R9-GFP using pP79 plasmid. The fluorescence of KrPL cultures was significantly higher when R9-GFP was produced (Appendix A), suggesting that this construct is characterized by improved properties in comparison to the canonical eGFP also at low temperatures.

#### 3.2.3. Influence of Medium Composition on pP79 Efficiency

We performed all our expression trials in GG, a defined medium whose only carbon sources are D-gluconate and L-glutamate [3]. The uptake of substrates is hierarchical in *Ph*TAC125 [7,28] and specific amino acid combinations had to be formulated in the past to guarantee the optimal induction of another psychrophilic expression plasmid in this bacterium [5]. To understand if this kind of interference could be experienced also using pP79, we measured the achievable levels of β-galactosidase production when KrPL pP79-*lacZ* was grown in TYP, a complex medium containing yeast extract and bacto-tryptone. The growth curves of the recombinant bacteria cultivated in GG and TYP at 15 °C are reported in Appendix A, respectively. As observable in Appendix A, the performance of the expression plasmid was similar in the two media when the lower IPTG concentration (1 mM) was reached in the cultures. On the other hand, at the two higher tested IPTG concentrations the β-galactosidase production was less efficient in the complex broth, indicating that in those conditions some negative effects took place.

### 3.3. Optimization of IPTG Transport Mechanism

#### 3.3.1. Attempts in the Plasmidic Expression of a Lactose Permease

For an in-depth study of the pP79 system, the influence of the IPTG transport mechanism was evaluated in relationship to protein expression. The requirement of a high concentration of IPTG (10 mM) to reach full induction and the absence of the lactose metabolic pathway in *Ph*TAC125 [29] suggest the potential absence of high-affinity IPTG transporters in this bacterium. If so, the mechanism by which the inducer penetrates within the cells is supposed to be either simple or facilitated diffusion.

This observation led us to examine whether the heterologous expression of a suitable lactose permease could deliver a significant contribution to the optimization of the pP79 system. Since the *E. coli* LacY transporter has already been successfully used in other Gram negative bacteria [30], its encoding gene was optimized for the codon usage of *Ph*TAC125 and cloned into pFC [21] and p13C, a pPM13 derivative [1]. These constitutive psychrophilic expression plasmids possess a medium and a strong promoter, respectively [1]. However, no transconjugant clones resulted from the mobilization of both plasmids into KrPL, probably due to toxic effects on the cell membrane deriving from the excessive production of the permease (data not shown). An alternative strategy was then applied through the integration of *E. coli lacY* gene into the genome of KrPL.

#### 3.3.2. Construction of KrPL *lon* and *lacY^+^* Mutant Strains

To ensure a subtoxic level of LacY in KrPL a mutant strain was developed so that the production of the permease derived from a single copy of *lacY*, integrated within the host genome. 

Firstly, we focused on the selection of the target gene for the integration of *lacY*. To obtain a mutant strain displaying improved features as a host for recombinant protein production, the centerpiece of our analysis was the set of genes coding for proteases that are constitutively expressed in *Ph*TAC125 and are involved in the proteolytic process of recombinant proteins. The Lon protease encoding gene (*PSHA_RS10175*) was chosen as the target of mutagenesis because it represents the major protein quality control protease and, as such, is responsible for most of the ATP-dependent degradation of misfolded proteins in bacteria [31]. Despite this protease is involved in a wide range of cellular functions (from proteins degradation to DNA replication and recombination, stress response, motility and biofilm formation), it is not an essential enzyme in many bacterial species such as *E.coli* [32]. To evaluate the consequence of *lon* disruption in *Ph*TAC125, a first mutant strain was constructed through a two-step integration–segregation approach using pVS, a suicide vector suitably constructed for *Ph*TAC125 [22]. Two internal gene fragments of *lon* were chosen as homologous sequences for the recombination event, amplified by PCR and cloned into pVS, resulting in pVS-*lon*. The first fragment (named A) is located into the sequence encoding the N-terminal domain of Lon, while the second fragment (named B) includes the sequence encoding the region straddling the ATPase domain and the proteolytic domain (Figure 5A). Since two different crossing-over events could occur, depending on which fragment underwent recombination, the two homologous regions were selected in order to provide the disruption of the whole Lon protease in the case of recombination of fragment A or the deletion of its proteolytic domain in the case of recombination of fragment B (Figure 5B). Once obtained, pVS-*lon* was mobilized into KrPL by intergeneric conjugation and a single recombination event allowed the vector insertion on the genome. PCR analyses demonstrated that the insertion occurred in fragment B of the gene (data not shown) resulting in a mutant that contains two non-functional copies of the *lon* gene. The first one encodes a truncated form of the Lon protease because it is devoid of the fragment B downstream sequence, coding for the active site domain of the protease. The second copy is not transcribed because it lacks its promoter and the 5′-encoding region (Figure 5C).

The presence of the truncated form of Lon was evaluated through Western blot analysis carried out on total KrPL cellular extracts with a polyclonal anti-Lon antiserum. As shown in Figure 6, the Lon band signal was detected at different heights in the wild-type and the mutant strains: the first one exhibited a band compatible with the expected molecular weight of the full-length protein (87.4 kDa, lane 1 in Figure 6), whereas the lower band of the mutant strain was clearly the truncated form (lane 2 in Figure 6). Indeed, the partial deletion of the first copy of the *lon* gene caused the loss of 564 bp at the 3′-region, generating a truncated protein with a theoretical size of 66 kDa.

The growth behavior and the fitness of *lon* mutant were then compared to the wt strain and no deleterious effects took place (Appendix A). Thus, *lon* was confirmed as the target of insertion of *EclacY* gene and the strain mutated in *lon* was used as its isogenic control. With this purpose, an expression cassette—consisting of the strong constitutive psychrophilic promoter P13 [1] and the *E. coli lacY* gene—was designed and included between two intragenic fragments of the *lon* gene. The entire construct was then cloned into pVS resulting in pVS-*lacY*. With a similar strategy used for the obtaining of the *lon* mutant strain, pVS-*lacY* was designed to obtain a truncated Lon protease devoid of its proteolytic domain. To do this, the fragment B, already used for the construction of KrPl *lon*, and the fragment B’, including the upstream region of the active site of Lon, were chosen as target sequences (Figure 7A). 

The obtained pVS-*lacY* was then mobilized into KrPL through interspecies conjugation and the genomes of the mutant clones were analyzed by PCR to define the presence and the orientation of the insertion (data not shown). Here too, the recombination event occurred in fragment B, succeeding in the disruption of *lon* and insertion of *lacY*. The obtained KrPL *lacY*^+^ mutant strain is potentially capable of producing a lactose permease and a truncated form of Lon protease (Figure 7C).

### 3.4. Comparison between the Performance of KrPL lon and KrPL lacY^+^ Strains

#### 3.4.1. Evaluation of the Production Improvement at Different Temperatures

To study whether the lactose permease provides an improvement to the transport of IPTG within the cell, the recombinant production of β-galactosidase was performed in *lacY^+^* in comparison to its isogenic control, *lon* mutant. Both strains were transformed with the expression vector p79C-*lacZ* and grown in GG medium at 15 °C. During the middle exponential growth phase, the induction was performed with different concentrations of IPTG. In particular, 0.05 mM, 0.1 mM, 0.5 mM, 1 mM IPTG was added to the culture to examine the difference in β-galactosidase production. The β-galactosidase activity was then assayed in the soluble cellular extracts recovered 8, 24, 32 and 48 h after the induction. As shown in Figure 8A, the highest production was achieved with the *lacY^+^* mutant and proved to be about 5-fold superior to *lon* strain. In the tested range of IPTG concentration, a direct proportionality between the inducer amount and production level was observed in both strains, but with a higher slope in *lacY^+^* mutant. As an example, Figure 8B highlights the linear correlation between β-galactosidase activity and IPTG concentration for the last data points. Bacterial cells containing the lactose permease yielded high levels of production both with 0.5 mM and 1 mM IPTG. However, the increase of inducer concentration in this range only drove a slight improvement in recombinant production. This is probably due to the decrease in the contribution of the lactose permease in the IPTG uptake when its concentration is relatively high, compatibly with its saturation. Furthermore, these results highlight that the minimum concentration of IPTG needed for the induction of expression in the strain containing the LacY transporter is 10-fold lower in comparison with the strain lacking the permease. When the induction of *lacZ* expression was performed with 0.05 mM and 0.1 mM IPTG in *lon* mutant, no difference in β-galactosidase activity is observed in comparison to the non-induced cells (NI). Hence, the lactose permease is a very important contributor in transporting IPTG across the *Ph*TAC125 membrane.

To verify that the mesophilic membrane protein is produced and functioning in the psychrophilic bacterium even at ultra-low temperatures, the recombinant production of β-galactosidase was performed at 0 °C using 0.5–1 mM IPTG. The levels of the reporter protein were then assayed after 24, 48 and 72 h from the induction. The specific activity of β-galactosidase measured in *lon* mutant highlights a poor accumulation of the protein, suggesting that the response of the system is owing to the basal expression of the protein (Figure 9). On the contrary, the effect of LacY in the transport of IPTG is noticeable already after 24 h of expression, with an enhancement of the production in *lacY^+^* strain of about 1.5-fold in comparison to *lon* cells treated with the same inducer concentration. As with the expression trials at 15 °C, 0.5 and 1 mM IPTG triggered the same expression levels in *lacY^+^* mutant also at 0 °C, except for the first time point where a higher recombinant production was guaranteed by increasing amounts of inducer.

#### 3.4.2. Evaluation of β-Galactosidase Production Using Lactose as an Inducer

A further demonstration of the functioning of *Ec*LacY transporter in the mutant strain was performed through the recombinant expression of *lacZ* by using 2% (*w/v*) lactose as an inducer. As shown in Figure 10A, a rapid increase in β-galactosidase activity was observed in *lacY^+^* strain after the first 4 h of expression, while no production was detected in *lon* mutant. Nonetheless, after 8 h of induction, the amount of recombinant protein took a decreasing trend. This can be traced back to a toxic effect observed when the cells can transport the lactose within the cells (Figure 10B). To better understand the reason for this occurrence, the growth behavior of the cells bearing pP79-*lacZ* and pP79 in the presence of lactose was compared. As reported in Figure 10B, only the cells capable of producing β-galactosidase showed cell death, suggesting that this effect was potentially caused by the metabolism and degradation of the disaccharide.

## 4. Discussion

The ability to produce heterologous proteins with high yields is a prerequisite for the exploitation of a microorganism as a cell factory [33]. The psychrophilic bacterium *Ph*TAC125 represents a model as a non-conventional host for the production of difficult to express proteins in a soluble and active form [2,5,34,35]. In the present paper, the set of plasmids for controlled gene expression in *Ph*TAC125 and KrPL—a pMtBL deficient strain—has been expanded with the IPTG-inducible plasmid pP79. Moreover, the usability of this plasmid for different purposes has been widened by the development of an engineered KrPL strain. All our experiments were carried out in the cured *Ph*TAC125 strain to avoid instability issues possibly arising from the coexistence of pMtBL and pP79 or pMAV.

In our selection of a new expression system, we looked for characterized psychrophilic genes involved in carbohydrate catabolism with a clear regulator-catabolic gene asset. In this sense, the choice of the *lacR-lacZ* gene couple of *Ph*TAE79 was immediate, considering the high levels of β-galactosidase produced by this bacterium [8] and the luck of *lacZ*-based inducible plasmids in other prokaryotes [33]. Rather than a disadvantage, the lack of lactose metabolism in the chosen host *Ph*TAC125 [29] can be seen as a possible prerequisite for a more predictable and tunable expression. As a matter of fact, the integration of a heterologous regulatory network in a new context can provide the basis to avoid undesired autocatalytic phenomena as the ones due to uneven and uncontrollable inducer transport [36] causing either bistable or “all-or-none” responses [37,38,39]. The main prerequisite needed for the functioning of *lacZ* induction in KrPL was the possibility of the internalization of its inducer. Both Hoynoux et al. [8] and our group demonstrated that IPTG could be used as a molecule regulating LacR activity. This reinforced our idea of implementing this recombinant system in KrPL, given the capability of this inducer to penetrate biological membranes in a diffusive manner also in transporter-deficient strains [30,36,40]. 

The pP79 vector proved to be more efficient than pMAV—a D-galactose inducible plasmid previously used in *Ph*TAC125 [3]. In particular, with the new system, we could accumulate a 20-fold higher quantity of β-galactosidase than pMAV-*lacZ* and we could detect the production of a fluorescent reporter for the first time in this bacterium. Moreover, we demonstrated that the growth broth composition had an impact on the levels of expression, i.e., a rich medium caused partial repression of the IPTG mediated induction of pP79. Understanding the underlying mechanisms of this negative regulation (e.g., inducer exclusion or repression of the regulator) will be important in the future to increase the extent of fine regulation that can be applied to pP79 [41,42]. However, it is worth noting that the deeply different formulations of the two used media gave rise to drastic different behaviors of the bacterial growths in general (Appendix A). In particular, the use of TYP guaranteed the doubling of the specific growth rate and the final biomass concentration, suggesting that diversified metabolic networks are activated in this “feast” condition. Hence, to dissect the processes that interfere with pP79 activity, slighter progressive modifications of the medium composition must be applied in the future.

Despite their promising features, KrPL pP79 recombinant strains needed high IPTG concentrations to reach maximal expression, a demand that could be prohibitive in large scale applications requiring induction levels as high as possible.

To approach this problem, we applied a strain engineering strategy based on the combination of the psychrophilic regulatory elements derived from *Ph*TAE79 with the mesophilic lactose permease LacY from *E. coli*. Despite the use of plasmids with a low copy number [26] and two different constitutive promoters with medium and high strength, the accumulation of the permease and the alteration of membrane properties probably caused serious toxic phenomena in *Ph*TAC125 [43]. This led to the failure of the first attempt of heterologous expression of LacY.

A further effort was made to modify the host cell to accommodate the production of the membrane protein through the integration of the *lacY* gene into the KrPL genome. Emphasis was concomitantly given to the control of the deleterious proteolysis of recombinant products with the final aim to design a more robust cell factory with improved features for various biotechnological applications. The novel mutant strain *lacY^+^,* constructed through a genome-scale manipulation, was characterized by a deletion in the proteolytic domain of Lon protease and the capability to produce the lactose transporter LacY in a functional form (Figure 7).

Previous analyses performed to characterize the truncated Lon protease suggest that it could act as a molecular chaperone [44]. Indeed, mutations in the active site abolish proteolysis but not ATPase activity, resulting in a protease that is still able to bind its substrates without degrading them [44]. The occurrence of the same phenomenon in KrPL mutant strains has to be proved by analyzing the production of more complex and unstable proteins than the ones studied in this work. 

The functional characterization of the *lacY^+^* mutant highlighted great differences in the levels of the reporter protein produced in comparison to its isogenic control, *lon* strain (Figure 8A). Despite no changes in the growth behavior and kinetics were observed between the two strains (Appendix A), a 5-fold increased protein accumulation was observed for *lacY^+^*, showing a higher slope of the direct proportionality between the production level and the inducer concentration. Owing to the cost and the possible cytotoxic effect of high concentrations of IPTG, this feature remarks the potentiality of *lacY^+^* strain as a cell factory given the advantage to use less inducer to reach the same level of recombinant production.

Furthermore, our novel mutant succeeded in the production of the mesophilic membrane protein in a functional form also at 0 °C, allowing the enhancement of the production levels of a reporter in comparison to its counterpart *lon* strain (Figure 9). This result is quite impressive, considering that the expressed lactose permease is naturally used to work at 37 °C and the known effects of the low temperatures on the membrane structure and composition [45].

Additional evidence to support the functionality of the transporter in *lacY^+^* strain was then gained by inducing the recombinant production of the β-galactosidase with lactose. This experiment confirmed that little or no lactose molecule penetrates inside *lon* cells by either other facilitated transport systems or diffusion and LacY is required for the disaccharide internalization (Figure 10A). However, a deleterious effect on cell viability was observed in *lacY^+^* cells, resembling a “lactose killing” phenomenon [46]. Surprisingly, the comparison of the growth behaviors of recombinant KrPL *lacY^+^* bearing pP79-*lacZ* and pP79 highlights that the cause of the observed stress is not related to the de-energization caused by elevated transmembrane lactose transport. Instead, a toxic accumulation of its catabolic products likely takes place (Figure 10B). As a matter of fact, the only difference between the two used strains consists of the production of the β-galactosidase which is causative of the conversion of lactose to catabolic intermediates. For this reason, these experiments carried out with lactose must be taken as proof of LacY functioning in KrPL, rather than as an example of its actual use for induction purposes. *Ph*TAE79 LacR is predicted to be an AraC-type protein and we experimentally demonstrated that its activity is regulated by IPTG, an allolactose analog. Considering that we used a β-galactosidase as a reporter and that allolactose is a product of the reaction catalyzed by this enzyme [47], it is very likely that part of the observed lactose-mediated induction is due to the peculiar activity of this protein (see Appendix B and Figure A1). Accordingly, when we used the same approach for the expression of other genes than *lacZ* in KrPL *lacY^+^* pP79, we could observe a lactose-mediated induction, but it was considerably lower than the one achievable with IPTG (data not shown), suggesting again that allolactose is probably the main inducer of this system. Our analysis of KrPL mutants demonstrated that LacY contributes to improving the recombinant production yield in *Ph*TAC125 when the *lacZ* promoter is employed. This finding is in agreement with most studies performed in E.coli and other Gram negative bacteria [30] and is noteworthy given that the IPTG uptake mechanism can be mediated by lactose permease, passive diffusion, or other types of permeases [40]. 

Altogether, our data certify that in both KrPL *lon* and KrPL *lacY^+^* strains an average graded production is possible upon IPTG induction (Figure 8B). Depending on the particular application, the two strains can be differently employed. As with the *E. coli* Tuner(DE3) strain (Novagen, [33]), either KrPL or KrPL *lon* could be successfully used for those studies requiring low production levels with a clear linear response over a wide IPTG concentration range, as in the case of metabolic engineering and of the production of toxic proteins. On the other hand, KrPL *lacY^+^* can be useful for those processes where an average graded response is still guaranteed, though not always linear (Figure 8B), and high production is triggered by a low concentration of the inducer [39]. This might be the case of the synthesis of non-problematic proteins and of low added value products requiring the containment of the production costs. However, one has to keep in mind that all our studies recorded the average expression levels of the cultures, and certain conclusions about the homogeneity of the induction cannot be deduced. Nevertheless, the introduction of the GFP as a reporter in *Ph*TAC125 for the first time opens to the possibility of single-cell studies as FACS screenings that can address this question.

Finally, although it is beyond the scope of this study, it is worth drawing some considerations about the selection of reporter genes to study promoter strengths at 15 °C. The *Ph*TAE79 β-galactosidase and R9-GFP were the main tools used to study pP79. Despite they both demonstrated a higher protein accumulation in pP79 recombinant strains than pMAV bearing cells, the production of the fluorescent reporter was way more inefficient than the β-galactosidase and this phenomenon was not related to transcriptional efficiency. More reasons can justify this difference. First, in pP79 the β-galactosidase encoding sequence is directly fused to its natural 5′ UTR, while the R9-GFP CDS is artificially joined with the psychrophilic 5′ UTR. Different groups have widely reported how the 5′ UTR composition and the fusion with heterologous translated sequences can cause translational issues [48,49,50]. In this sense, even the comparison of pMAV and pP79 relative strengths might be partially biased by the fact that they harbor the galT and *lacZ* 5′ UTR sequences, respectively.

Nevertheless, in the past a similar disparity in terms of protein accumulation was also observed when the production of the psychrophilic β-galactosidase and a mesophilic α-glucosidase were compared in recombinant *Ph*TAC125 pUCRP strains [2]. Also, in that case the synthesis of the cold-adapted enzyme was higher that the mesophilic one, indicating that even when fused to a heterologous 5′ UTR, the *lacZ* CDS is efficiently translated. Hence, the remarkable accumulation of *Ph*TAE79 β-galactosidase may be strictly related to either a high translation efficiency or protein stability at low temperatures. This hypothesis is corroborated by a study effectively demonstrating that the *Ph*TAE79 β-galactosidase showed a particularly high activity when produced in a heterologous bacterium at low temperatures, despite the low levels of transcription [51]. Collectively, these observations may suggest that even if *Ph*TAE79 *lacZ* can be a useful tool for comparative studies, as done in this work, it might not reliably represent the absolute achievable amounts of other proteins at low temperatures using the same system, especially in the case of mesophilic proteins.

On the other hand, the production of fluorescent reporters in KrPL was so low not to allow the detection of the expression starting from weak promoters, as the one of pMAV vector. Originally, we used an R9-GFP variant, possessing an N-terminal oligoarginine peptide and the mutations characteristic of the eGFP [19] and of Cycle 3 GFP [27]. We intended to combine the enhanced fluorescence of eGFP and improved folding features of Cycle 3 mutations to the aggregation-prone polyarginine peptide [18,52]. In this way, we wanted to confer an increased in vivo stability to the protein, typical of full-protein nanoparticles [53]. However, all these modifications were predicted to improve the GFP properties at 37 °C with no further information about their consequences at lower temperatures. That is why we also produced a plain codon optimized eGFP—called pGFP—with neither R9 nor Cycle 3 mutations. The R9-gfp gene has a Codon Adaptation Index (CAI) of 0.66, while the *pGFP* gene has a CAI equal to 0.73 [20], but R9-GFP derived fluorescence was considerably higher. This indicates that at least some of the modifications introduced in R9-GFP had a positive effect. Nevertheless, the actual necessity to use the R9 peptide has still to be tested, considering the translational and degradation issues possibly deriving from N-terminal oligoarginines [50,54] and that GFP nanoparticles can sometimes show worsened fluorescent properties [55]. To make this fluorescent system more sensitive, other approaches have to be pursued, such as the shift of the emission spectrum to a wavelength for which *Ph*TAC125 experiences a lower autofluorescence and the application of protein mutations that improve the fluorophore maturation at low temperatures. However, both pGFP and R9-GFP are already detectable in combination with pP79, allowing for new kinds of study.

## 5. Conclusions

There is still a significant number of predicted protein products whose recombinant production in conventional gene expression systems is unsuccessful, making their structural/functional characterization and their biotechnological application impossible. Almost 20 years ago, our research group suggested the use of *Ph*TAC125 and its derived genetic tools for the setup of a novel cell factory working at low temperatures [14]. Till then, much evidence highlighted the notable skills of the Antarctic bacterium in the high quality production of human and/or eukaryotic complex proteins, reinforcing our original idea [2,3,5,6,11,34,35].

In this paper, we achieved a further considerable improvement toward the actual application of *Ph*TAC125 as an industrial cell factory. Based on the study of the regulated genetic elements in the psychrophilic bacterium *Ph*TAE79, we developed pP79, a novel IPTG-inducible plasmid. By using this expression system, we obtained about 20-fold higher production of the recombinant β-galactosidase in comparison to pMAV, the previous best inducible genetic system exploited in *Ph*TAC125 [3]. For the first time, the detection of a fluorescent protein was achieved in *Ph*TAC125 pP79 recombinant cells, paving the way for a variety of sensitive and innovative approaches of study.

Another essential aim of this work was to demonstrate the feasibility of a rational approach toward the host improvement. The inducer internalization and the control of proteolytic events were addressed, constructing the engineered strain *lacY^+^* capable of producing a mesophilic lactose permease and a truncated form of Lon protease. This mutant strain allowed a 5-fold higher production than its isogenic *lon* mutant using a lower IPTG concentration. Furthermore, the heterologous permease showed its positive contribution to induction at 0 °C, widening the applicability of KrPL *lacY^+^* also as a host for the recombinant protein production at ultra-low temperatures.

## Figures and Tables

**Figure 1 microorganisms-08-01466-f001:**
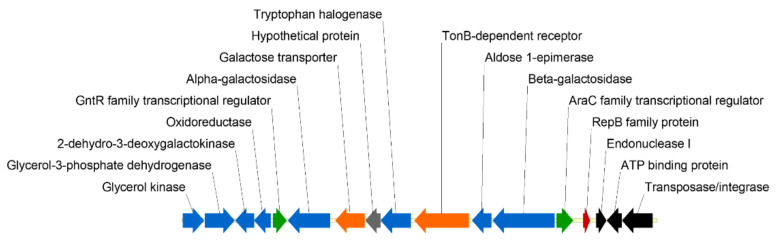
Disposition of genes surrounding the β-galactosidase encoding gene in *Pseudoalteromonas haloplanktis* TAE79 (*Ph*TAE79) AUTL01000130.1 contig. Blue arrows indicate genes involved in metabolism, in orange receptors and transporters, while in green transcriptional regulators. Black arrows depict genes involved in DNA rearrangements, the red arrow highlights the presence of a gene involved in plasmid replication and the gray arrow indicates a coding sequence with unknown functions.

**Figure 2 microorganisms-08-01466-f002:**
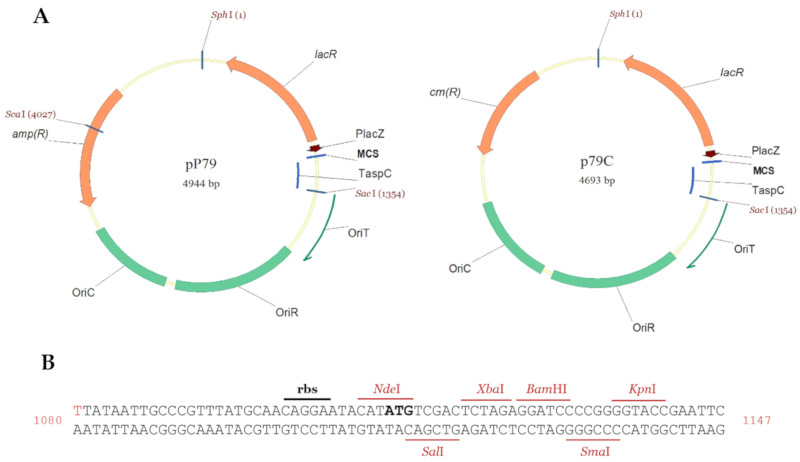
Maps of pP79 and p79C shuttle vectors. (**A**) The two plasmids differ for the selection markers, which are a β-lactamase encoding gene (*amp*(R)) in pP79 and a chloramphenicol acetyltransferase encoding sequence (*cm*(R)) in p79C. The common elements to the plasmids are *Ph*TAE79 regulatory gene *lacR*, the promoter of *Ph*TAE79 *lacZ* gene (P*lacZ*), the transcriptional terminator on *Ph*TAC125 *aspC* gene (T*aspC*), an origin of conjugative transfer (OriT), the pMtBL-derived replication origin for the maintenance in *Ph*TAC125 (OriR) and the pUC18-derived replication origin for the propagation in *E. coli* (OriC). Restriction sites outside the MCS that have been used for cloning purposes are indicated. (**B**) The sequence encompassing the 5′ UTR and the MCS of the two plasmids included between P*lacZ* and T*aspC* is illustrated using the coordinates of pP79. The +1 of the mRNA and the start ATG are indicated in red and bold black, respectively.

**Figure 3 microorganisms-08-01466-f003:**
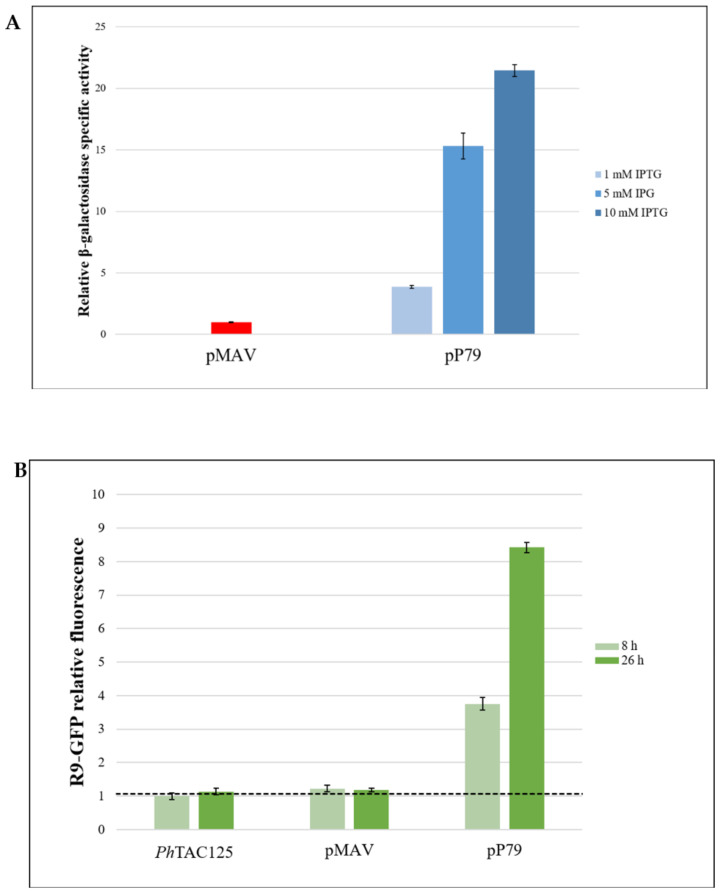
Quantification of the relative strengths of pMAV and pP79-driven expression of *lacZ* and *R9-gfp* genes at 15 °C. (**A**) β-galactosidase production measured with an enzymatic assay. Logarithmic cultures of KrPL pMAV-*lacZ* and KrPL pP79-*lacZ* strains were exposed to 10 mM D-galactose and 1-10 mM IPTG, respectively. After 26 h expression, the β-galactosidase activities were assayed. The enzymatic specific activities are reported as measures normalized by pMAV-*lacZ*. (**B**) R9-GFP synthesis was triggered with 10 mM D-galactose in the case of pMAV-*R9-gfp* bearing strain and with 10 mM IPTG in the case of KrPL pP79-*R9-gfp*. The recorded fluorescence intensities were scaled to the autofluorescence of wild-type cells (*Ph*TAC125 bars and horizontal dashed line). Levels of β-galactosidase activity and fluorescence are expressed as mean ± SD, *n* = 3.

**Figure 4 microorganisms-08-01466-f004:**
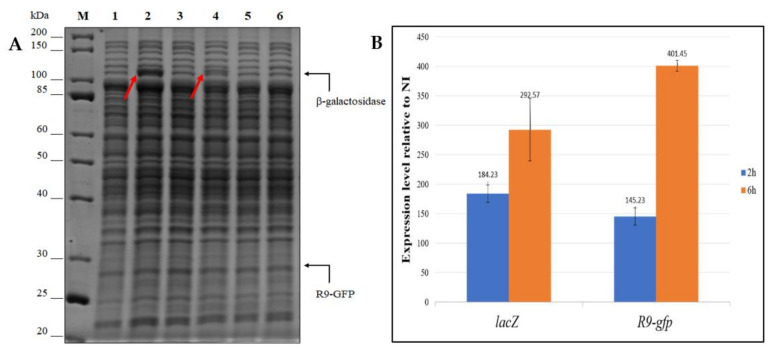
β-galactosidase and R9-GFP proteins productions (panel **A**) and their respective mRNA transcriptions (panel **B**). (**A**) SDS-PAGE analysis of cell extracts of KrPL strains producing the β-galactosidase (lanes 1–4) and R9-GFP (lanes 5–6). The induction was performed with 10 mM D-galactose for pMAV carrying strains and with 1 mM IPTG for pP79 bearing cells and protracted for 26 h. M, molecular weight marker; 1, non-induced pP79-*lacZ*; 2, induced pP79-*lacZ*; 3, non-induced pMAV-*lacZ*; 4, induced pMAV-*lacZ*; 5, induced pP79-*R9-gfp*; 6, induced pMAV-*R9-gfp*. Black arrows on the right of the gel represent the expected molecular weights of the recombinant proteins. Red arrows inside the gel highlight the bands of the β-galactosidase. (**B**) Relative quantification by RT-qPCR of mRNA expression levels of *lacZ* and *R9-gfp*. Two genes under the control of the PlacZ promoter were analyzed for their mRNA expression levels after 2 and 6 h from the induction in comparison to the non-induced condition. The reported results are the mean of three independent experiments.

**Figure 5 microorganisms-08-01466-f005:**
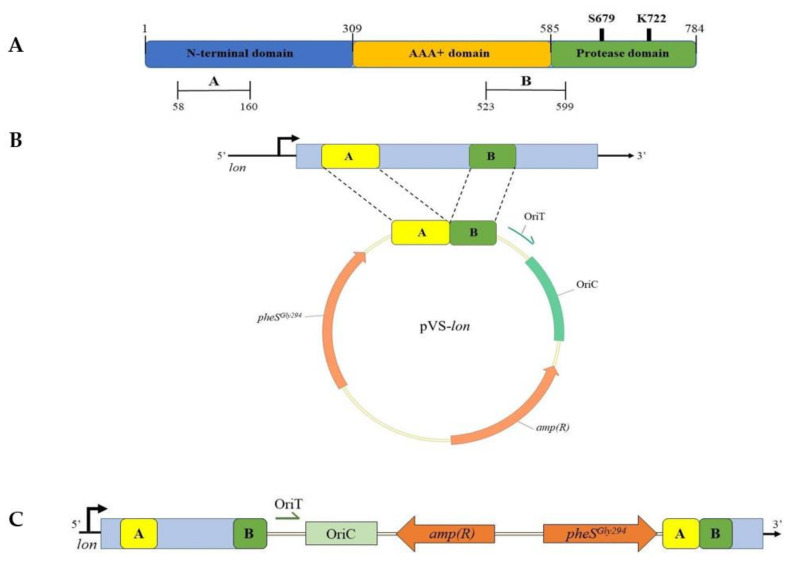
Construction of KrPL *lon* mutant strain. (**A**) Domain organization of Lon protease. The N-domain is involved in substrate recognition and binding; the AAA+ domain contains the ATPase module; the Protease domain is responsible for proteins degradation. S679 and K722 represent the catalytic dyad of the proteolytic domain. Fragments A and B encoding sequences were chosen as recombination regions. (**B**) Schematic representation of pVS-*lon* vector. (**C**) Genetic organization of KrPL *lon* selected mutant.

**Figure 6 microorganisms-08-01466-f006:**
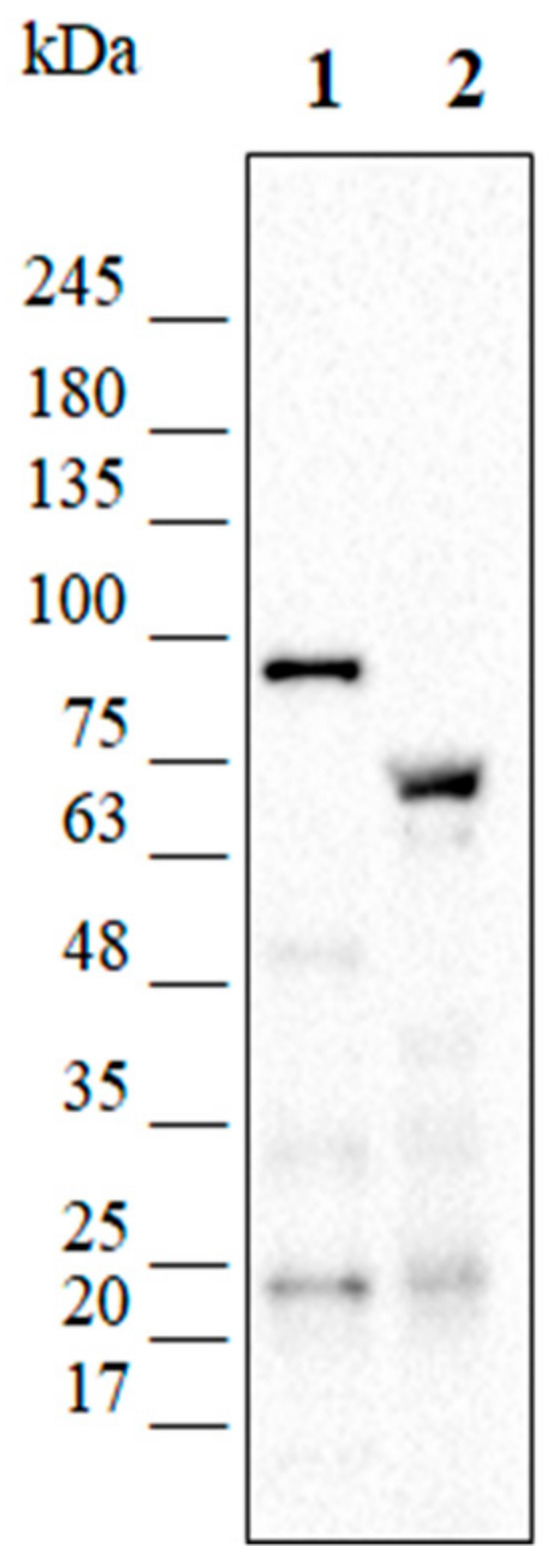
Western blot analysis carried out with anti-Lon antiserum. Total cellular extracts of KrPL wt and *lon* mutant were analyzed through Western blot analysis. Lane 1 shows a signal corresponding to the full-length form of Lon protease (expected size 87.4 kDa) in the wt strain. A lower band (theoretical size of 66 kDa) is detected in the selected *lon* mutant strain corresponding to the truncated form of the protein.

**Figure 7 microorganisms-08-01466-f007:**
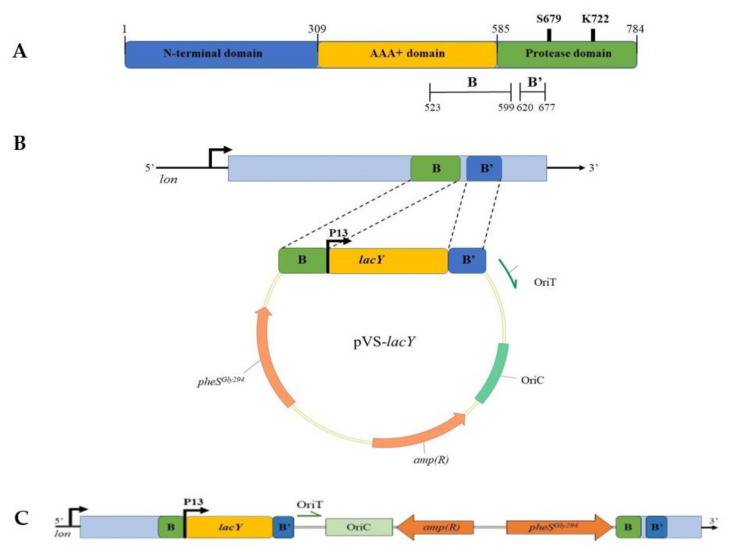
Construction of KrPL *lacY*^+^ mutant strain. (**A**) Domain organization of Lon protease. The N-domain is involved in substrate recognition and binding; the AAA+ domain contains the ATPase module; the Protease domain is responsible for proteins degradation. S679 and K722 represent the catalytic dyad of the proteolytic domain. Fragments B and B’ encoding sequences were chosen as recombination regions. (**B**) Schematic representation of pVS-*lacY* vector. (**C**) Genetic organization of KrPL *lacY*^+^ selected mutant.

**Figure 8 microorganisms-08-01466-f008:**
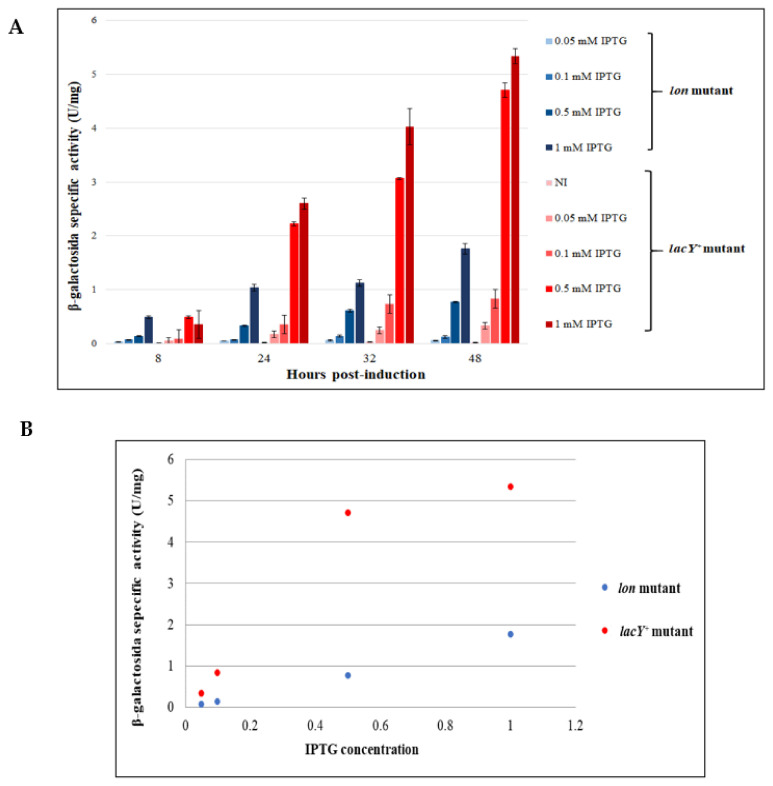
Evaluation of production performance of KrPL *lon* and *lacY*^+^ mutant strains. (**A**) β-galactosidase specific activity (U/mg) in *lacY*^+^ and *lon* mutant cells harboring p79C-*lacZ*, collected after progressive times of induction, in GG medium at 15 °C using different concentrations of inducer. Levels of β-galactosidase activity are expressed as mean ± SD, *n* = 3. (**B**) Analysis of the relationship between IPTG concentrations and β-galactosidase specific activity measured after 48 h of expression in *lacY*^+^ and *lon* strains.

**Figure 9 microorganisms-08-01466-f009:**
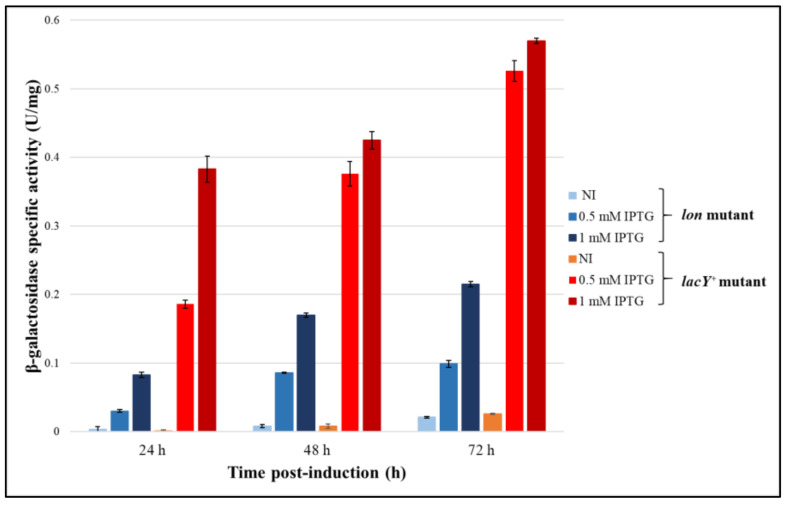
Evaluation of production performance of KrPL *lon* and *lacY*^+^ mutant strains at 0 °C. β-galactosidase specific activity (U/mg) in *lacY*^+^ and *lon* mutant cells harboring p79C-*lacZ*, collected after progressive times of induction, in GG medium at 0 °C using different concentrations of inducer. Levels of β-galactosidase activity are expressed as mean ± SD, *n* = 3.

**Figure 10 microorganisms-08-01466-f010:**
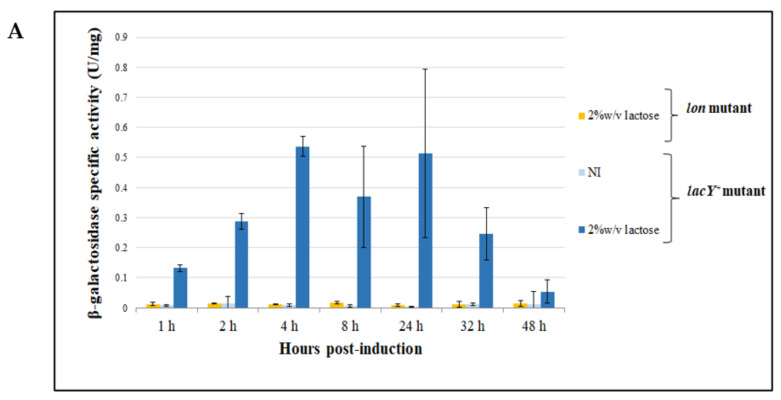
(**A**) Evaluation of β-galactosidase production in KrPL *lacY*^+^ strain using lactose as the inducer. β-galactosidase specific activity (U/mg) in *lacY*^+^ and *lon* mutant cells harboring p79C-*lacZ* collected after progressive times of induction in GG medium at 15 °C using 2% (*w/v*) lactose as inducer. Levels of β-galactosidase activity are expressed as mean ± SD, *n* = 3. (**B**) Growth curves of KrPL *lacY*^+^ harboring p79C-*lacZ* and pP79 in the presence of 2% (*w/v*) lactose. The growth was performed at 15 °C in GG medium. The moment of the induction is represented by the intersection of the axes. The measures of optical density are expressed as mean ± SD, *n* = 2.

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
