# Peer review of "Improvement of Pseudoalteromonas haloplanktis TAC125 as a Cell Factory: IPTG-Inducible Plasmid Construction and Strain Engineering"

_microorganisms, 2020, doi:10.3390/microorganisms8101466_

Round 1

Reviewer 1 Report

The research work reported in this manuscript is aiming at improving protein expression in Pseudoalteromonas haloplanktis by designing a new plasmid inducible by IPTG. This includes improving inducer transport inside the cell (lactose permease) and generating a truncated form of Lon protease.

The article is well organized and written and the discussion is clear. I have only details remarks.

Line 85: Please correct the region numbering

Lines 154 and 156: two couples. Please add information on DNA polymerase used for the amplification

Line 231: Maybe, it is better to use 'horizontal gene transfer' instead of 'plasmidic origin'

Line 274: Why was this strain cured and this plasmid removed? Maybe you can explain a little more.

Line 654: a-glucosidase should be corrected (greek symbol)

In the bibliographic references, Pseudoalteromonas haloplanktis should be written in italics and 'haloplanktis' first letter should not be upper case.

Author Response

Dear Reviewer 1,

thanks for your evaluation and suggestions. You'll find below our actions in reply to your requests.

R1: Line 85: Please correct the region numbering

>>>The coordinates have been corrected: 18’782-19’909.

R1: Lines 154 and 156: two couples. Please add information on DNA polymerase used for the amplification

>>>The used enzyme is now indicated, and the typo has been corrected

R1: Line 231: Maybe, it is better to use 'horizontal gene transfer' instead of 'plasmidic origin'

>>>The sentence has been changed as suggested by the reviewer.

R1: Line 274: Why was this strain cured and this plasmid removed? Maybe you can explain a little more.

>>>We thank the reviewer for the observation. Although we have not explored the possible interference of pMtBL on the recombinant plasmids, we decided to use a cured strain just to prevent any instability issue possibly arising in PhTAC125. Now this concept is expressed in the Discussion section (Lines 543 -547).

R1: Line 654: a-glucosidase should be corrected (greek symbol)

>>>The proposed correction has been made.

R1: In the bibliographic references, Pseudoalteromonas haloplanktis should be written in italics and 'haloplanktis' first letter should not be upper case.

>>>All the references have been changed using this formulation.

Reviewer 2 Report

In this manuscript, Colarusso and colleagues have proposed an improvement of Pseudoalteromonas haloplanktis  TAC125 as a cell factory utilizing a IPTG-inducible Plasmid 3 construction and strain engineering. The authors utilized appropriate technique of characterization and analysis (although some preparations of the samples for the analysis are missed), supplying useful information. There are some typing errors to correct during the revision of the work. Moreover some sentences are too complex and rambling. The changes and suggestions are listed below:

  1. Please clarify the following sentence: “The L-malate inducible pUCRP plasmid guaranteed a remarkable protein accumulation [2], but its efficacy resulted to be strongly influenced by the medium composition so that ad hoc broth formulation and fermentation strategies had to be contemplated in the scale-up design.
  2. Increase the quality of figure 1 and 2
  3. Please specify what kind of technique of homogenization you utilized for the preparation of the samples for SDS-PAGE. It is not clear if it is the same for the samples utilize for spectroscopy.
  4. Revise overall the manuscript for the presence of several typing and grammar errors.
  5. It is useful to fortify the “Conclusions” section to better explain the potentiality of the work

Author Response

Dear Reviewer 2,

following your evaluation and suggestions, we revised the manuscript to correct the typos errors, and the text was revised to make it more clear and readable. You'll find below our actions in reply to your requests.

R2: Please clarify the following sentence: “The L-malate inducible pUCRP plasmid guaranteed a remarkable protein accumulation [2], but its efficacy resulted to be strongly influenced by the medium composition so that ad hoc broth formulation and fermentation strategies had to be contemplated in the scale-up design.

>>> More details have been added (Lines 38-42). Although the pUCRP repression triggered by some amino acids has been successfully used for a tighter regulation in the past, its use could be more troublesome for general use purposes.

R2: Increase the quality of figure 1 and 2

>>> The overall quality of the figures has been increased.

R2: Please specify what kind of technique of homogenization you utilized for the preparation of the samples for SDS-PAGE. It is not clear if it is the same for the samples utilize for spectroscopy.

>>> A more detailed procedure is now described in Lines 192 -197.

R2: Revise overall the manuscript for the presence of several typing and grammar errors.

>>> Following the reviewer advice, we revised the whole manuscript to improve the understandability of the text.

R2: It is useful to fortify the “Conclusions” section to better explain the potentiality of the work.

>>> The Conclusions section was deeply revised. We hope that now it will clearly describe the impact of our results.

Other notes:

Table S2 has been updated with 4 primers mentioned in the text